# Hydrogen Sulfide Metabolism and Pulmonary Hypertension

**DOI:** 10.3390/cells10061477

**Published:** 2021-06-12

**Authors:** Lukas Roubenne, Roger Marthan, Bruno Le Grand, Christelle Guibert

**Affiliations:** 1INSERM, Centre de Recherche Cardio-Thoracique de Bordeaux, U1045, Avenue du Haut-Lévêque, F-33604 Pessac, France; lukas.roubenne@u-bordeaux.fr (L.R.); Roger.marthan@u-bordeaux.fr (R.M.); 2Centre de Recherche Cardio-Thoracique de Bordeaux, Univ Bordeaux, U1045, 146 Rue Léo Saignat, F-33000 Bordeaux, France; 3OP2 Drugs, Avenue du Haut Lévêque, F-33604 Pessac, France; bruno@op2drugs.com; 4CHU de Bordeaux, Avenue du Haut Lévêque, F-33604 Pessac, France

**Keywords:** pulmonary hypertension, hydrogen sulfide, vascular remodeling, vascular reactivity, oxidative stress, inflammation

## Abstract

Pulmonary hypertension (PH) is a severe and multifactorial disease characterized by a progressive elevation of pulmonary arterial resistance and pressure due to remodeling, inflammation, oxidative stress, and vasoreactive alterations of pulmonary arteries (PAs). Currently, the etiology of these pathological features is not clearly understood and, therefore, no curative treatment is available. Since the 1990s, hydrogen sulfide (H_2_S) has been described as the third gasotransmitter with plethoric regulatory functions in cardiovascular tissues, especially in pulmonary circulation. Alteration in H_2_S biogenesis has been associated with the hallmarks of PH. H_2_S is also involved in pulmonary vascular cell homeostasis via the regulation of hypoxia response and mitochondrial bioenergetics, which are critical phenomena affected during the development of PH. In addition, H_2_S modulates ATP-sensitive K^+^ channel (K_ATP_) activity, and is associated with PA relaxation. In vitro or in vivo H_2_S supplementation exerts antioxidative and anti-inflammatory properties, and reduces PA remodeling. Altogether, current findings suggest that H_2_S promotes protective effects against PH, and could be a relevant target for a new therapeutic strategy, using attractive H_2_S-releasing molecules. Thus, the present review discusses the involvement and dysregulation of H_2_S metabolism in pulmonary circulation pathophysiology.

## 1. Introduction

Pulmonary hypertension (PH) is a group of multifactorial and devastating cardiovascular conditions characterized by a progressive elevation of pulmonary arterial resistance (≥ 3 Wood) and mean pulmonary artery pressure (mPAP) above 20 mmHg [1], leading to right ventricular (RV) hypertrophy, failure, and ultimately to premature death [2,3]. PH is divided into five groups according to clinical, hemodynamic, and etiological characteristics, as well as treatment strategy: idiopathic and heritable pulmonary arterial hypertension (PAH) (group 1); PH due to left heart disease (group 2); PH due to lung diseases and/or hypoxia (group 3); PH associated with chronic thromboembolism (group 4); and finally, PH forms with unclear or multifaceted origins (group 5) [2]. PH development is associated with complex functional and structural modifications of the pulmonary arteries (PA), involving both pulmonary arterial endothelial cells (PAECs) and smooth muscle cells (PASMCs) [4]. Impairment of PA reactivity is a critical PH hallmark, and is linked to an imbalance of production and bioavailability of vasocontractile and vasorelaxant mediators [5,6]. This dysregulation is intimately linked to endothelial dysfunction [6,7], and is characterized by excessive constriction and decreased relaxation of PAs [5,8]. Concurrently, architectural changes of PA are mainly due to arterial wall remodeling, including media thickening of proximal arteries, intimal neo-formation, and fibrosis [4]. It is now well established that such changes are associated with PASMC and PAEC alterations during PH—namely, a pro-proliferative and apoptosis-resistant phenotype; an increased migration potential; extracellular matrix remodeling, mainly through excessive collagen deposition; and also endothelial-to-mesenchymal transdifferentiation [4,9]. Exacerbated inflammatory vascular responses also play a critical role in the development of PH. Actually, circulating concentrations of several proinflammatory cytokines, such as interleukin 6 (IL-6) and tumor necrosis factor alpha (TNF-α) [4,10], are abnormally increased in PAs regardless of the form of PH. This inflammatory context is associated with a worse clinical outcome in PH patients, and could contribute to exaggerated reactivity and remodeling of vascular cells. Furthermore, oxidative stress and associated enzymatic and/or mitochondrial reactive oxygen species (ROS) production are also considered to be important effectors in the development of PH, especially during PAs’ responses to hypoxia [11,12,13]. Currently, therapeutic strategies developed against PH are based on the use of a treatment targeted to reduce exacerbated pulmonary vascular contractility and remodeling, along with a more general treatment to alleviate respiratory symptoms and RV failure [2]. However, these treatments fail to cure PH, and lung transplantation remains too often a requisite solution for eligible patients with severe forms of PH resistant to medical management. In the absence of suitable treatments, patients with idiopathic pulmonary arterial hypertension have a mean survival of 2.8 years [2]. As a consequence, the search for new therapeutic targets in the landscape of PH’s cellular and molecular mechanisms remains essential.

Sulfur-based gases, such as hydrogen sulfide (H_2_S), have been known for a long time as environmentally poisonous and malodorous gases [14]. Aside from this harmful aspect, H_2_S is also involved in the regulation of several biological processes, and is currently considered to be a novel complementary gasotransmitter to the well-known nitric monoxide (NO) and carbon monoxide (CO). The growing importance of H_2_S in biological research is evidenced by the substantial increase in the number of original and review articles in the PubMed database dealing with this sulfur-based gas since 1996 (Figure 1). Like NO and CO, H_2_S exhibits high solubility in aqueous solution and in lipids, suggesting an efficient ability to cross plasma membranes [15]. H_2_S is enzymatically produced under various physiological conditions, and acts as an endogenous signaling molecule that does not require a second messenger or any receptor activation [15,16,17,18]. Cystathionine-β synthase (CBS) and cystathionine-γ lyase (CSE)—two pyridoxal 5′-phosphate (PLP)-dependent enzymes—are considered to be the main sources of H_2_S in mammalian cells [19,20,21]. A third enzyme—3-mercaptopyruvate sulfur transferase (3-MST)—has also been shown to release H_2_S along with cysteine aminotransferase (CAT) activity [21,22,23,24]. H_2_S was initially described as a vasodilator, lowering blood pressure [25,26] through the ATP-sensitive potassium channel (K_ATP_) openings in smooth muscle cells [27]. H_2_S also plays a critical role in numerous cardiovascular diseases, including atherosclerosis, diabetes-associated vascular disorders, and systemic hypertension [28,29,30,31,32], in which alterations of endogenous sulfur homeostasis can be observed [33].

In the context of the pulmonary circulation, several studies have already highlighted the essential role of endogenous H_2_S metabolism in both regulating physiological functions and contributing to the pathological hallmarks of PH [34,35,36,37]. CBS, CSE, and 3-MST are expressed in the intima and media of PAs [27,38,39,40,41,42,43,44], where H_2_S is supposed to regulate oxygen sensing and associated hypoxic pulmonary vasoconstriction (HPV) [38,39,45,46]. H_2_S activates smooth muscle cell K_ATP_ channels and, thus, contributes to PA relaxation [42], possibly in synergy with NO [47]. Interestingly, in vivo PH models are associated with dysregulated plasma H_2_S content and lung CSE expression [34,48,49,50]. Moreover, oxidative and endoplasmic reticulum (ER) stress are attenuated by exogenous H_2_S supplementation in several PH models [51,52,53]. Such dysregulation of H_2_S metabolism in various cardiovascular diseases, including PH, leads to the consideration of new therapeutic strategies based on natural or synthetic H_2_S-releasing molecules such as GYY4137, the dithiolthione family, and garlic-derived donors [26,54,55], which have already exhibited interesting effects on pulmonary circulation pathophysiology [51,52,56,57]. Thus, the present review focuses on recent advances in the involvement of H_2_S metabolism and the dysregulation of its production in pulmonary circulation, and their consequences in the context of PH. Future prospects for the potential use of H_2_S donors as new therapeutic tools are also discussed.

## 2. H_2_S Metabolism in the Pulmonary Vasculature

H_2_S is endogenously produced under physiological conditions mainly through enzymatic pathways, but several findings suggest a role of non-enzymatic pathways in H_2_S mobilization [58,59,60,61]. In a physiological context, 28% of H_2_S is in the undissociated form, 72% is dissociated into hydrosulfide anions (HS^-^), and the rest is sulfide anions (S^2−^), although in negligible amounts [62]. The effects of the undissociated and dissociated forms cannot be distinguished, and are grouped in the nomenclature H_2_S. Initial reports suggested that H_2_S concentration was approximately 30 μM in lung tissue, and rose to more than 200 μM in the heart, brain, and plasma [63]. However, methodological considerations related to variability in pH, temperature, substrates concentration, etc., suggested that these concentrations were far from physiological relevance [64]. In fact, steady-state H_2_S concentration was reported to be in the nanomolar range in most tissues and in plasma [63,64]. This range is consistent with the fact that H_2_S undergoes a high turnover rate in physiological conditions, characterized by an important production associated with rapid oxidation [64]. At high concentrations ((H_2_S) > 20 μM), H_2_S triggers detrimental cellular consequences [65], characterized by mitochondrial poisoning due to inhibition of cytochrome c oxidase (complex IV) [33,66,67]. Thus, the duality of H_2_S’s effects requires a fine regulation of the production/clearance balance.

### 2.1. Anabolic Pathways of H_2_S

In mammalian cells, H_2_S is primarily produced through the action of specific enzymes involved in homocysteine metabolism and L-cysteine catabolism: CSE, CBS, and 3-MST/CAT coupling. CSE was initially predominantly found in cardiovascular tissues, with marked expression in the heart and vessels [15,28]. CBS is mainly present in brain cells and the central nervous system although anterior reports suggest that when homocysteine levels are high, H_2_S can be produced by CBS in endothelial cells [21,68]. A more recent study demonstrated that the loss of CBS in endothelial cells decreases H_2_S production by 50% and dysregulates endothelial signaling [68]. CSE and CBS are two PLP-dependent enzymes mainly localized in the cytosol. Interestingly, several reports showed a possible translocation of CSE and CBS into mitochondria under cellular stress conditions [69,70]. In vascular smooth muscle cells, translocation of CSE from the cytosol to the mitochondria is induced by cytosolic calcium increase and, thus, promotes H_2_S production within the mitochondria, where L-cysteine concentration is three times higher than in cytosol [70]. Likewise, CBS is transferred into the mitochondria in response to hypoxia [69]. CSE and CBS have been considered for a long time to be the main endogenous H_2_S enzymatic sources in cells [19,20,21] by acting on L-cysteine, their predominant substrate. L-cysteine is a sulfhydryl amino acid derived from nutrition or endogenously generated from homocysteine via the canonical transsulfuration pathway, involving CSE and CBS (Figure 2a). Briefly, CBS converts homocysteine and L-serine into cystathionine, which is then metabolized by CSE in L-cysteine and α-ketobutyrate [71]. To generate H_2_S from L-cysteine, CSE and CBS catalyze a desulfhydration reaction and generate pyruvate and L-serine, respectively. Under physiological conditions, intracellular L-cysteine amounts exceed those of homocysteine, suggesting that L-cysteine desulfhydration is the main pathway of CSE-associated H_2_S production. However, in hyperhomocysteinemia, CSE can catalyze H_2_S release from two molecules of homocysteine [21,71], suggesting that the latter constitutes an alternative substrate for H_2_S production. It should be noted that H_2_S production by CBS is more efficient when homocysteine acts as a co-substrate of L-cysteine [71]. CBS was also shown to be allosterically activated and stabilized by S-adenosyl-L-methionine—a methyl group donor [71]. On the other hand, CO and NO inhibit CBS activity, highlighting gasotransmitter crosstalk [71]. CSE expression is upregulated by multiple factors—namely, ROS [72], ER stress [73], and calcium increase [74,75]. CSE transcription is also upregulated by the action of TNF-α during inflammation [76]. CSE activity is enhanced by calmodulin in the presence of high calcium concentrations [28], although this finding remains controversial [71]. Contrary to CBS, CSE expression and activity are stimulated by NO in vascular tissues using NO donors [27].

PLP-independent 3-MST localization in tissue was initially mentioned in brain tissue [23], and later in liver, large intestine, and kidney tissues [77]. However, 3-MST is also expressed in the endothelium and media of the thoracic aorta [22]. Cytosolic CAT1 and mitochondrial CAT2 are widely expressed in liver, heart, and kidney tissues [78]. CAT1/2 were also detected in the thoracic aorta, but only in endothelial cells [22], leading to a proposed endothelial predominance of H_2_S production by CAT/3-MST coupling. 3-MST is mainly present within the mitochondrial matrix, but is also detectable in the cytosol [66,79]. To produce H_2_S, 3-MST requires the activity of CAT, which exists in two forms: CAT1 and CAT2 localized in the cytosol and the mitochondria, respectively [21]. Indeed, CAT activity catalyzes transamination between L-cysteine and α-ketoglutarate in order to produce 3-mercatopyruvate. The latter is then used as a substrate by 3-MST to release H_2_S in presence of reducing molecules, such as endogenous thioredoxin (Trx) or dihydrolipoic acid (DHLA) [21,22,23,24]. Trx is a class of ubiquitously expressed redox proteins, including cytosolic Trx1 and mitochondrial Trx2 [24]. DHLA is a reducing compound present in the mitochondria [24]. Although 3-MST-associated H_2_S release occurs in the cytosol and the mitochondria, the steady-state contribution of mitochondrial and cytosolic 3-MST to cellular H_2_S production remains to be determined. Due to its predominant mitochondrial localization, H_2_S production by 3-MST contributes to mitochondrial metabolism through oxidation by sulfide quinone oxidoreductase (SQR), which is part of the mitochondrial electron transport chain (ETC) [80] (Figure 2b). 3-MST expression is also enhanced in human umbilical vein endothelial cells in response to shear flow [81]. In contrast, 3-MST activity can be inhibited by oxidative stress through redox modifications of its molecular structure [82].

Recent advances in endogenous H_2_S metabolism study suggest that H_2_S can be generated independently of CSE, CBS, and 3-MST/CAT coupling. Although minor, non-enzymatic H_2_S release occurs during the chemical reduction of reactive sulfur groups in thiosulfates or polysulfides (Figure 2a). Indeed, Benavides et al. demonstrated that dietary garlic (*Allium sativum*)-derived organic polysulfides (diallyl disulfide and diallyl trisulfide) can release H_2_S in presence of GSH and glucose metabolism [58]. In addition to being a substrate for H_2_S-generating enzymes, L-cysteine also generates H_2_S in physiological conditions in blood, in the presence of iron and vitamin B6 [59]. As a consequence, it has been suggested that basal circulating H_2_S content is due to this novel non-enzymatic process.

### 2.2. Catabolic Pathways of H_2_S

Control of H_2_S metabolism requires an efficient clearance in order to avoid its detrimental effects on mitochondrial function [66]. Indeed, at high concentrations, H_2_S inhibits oxygen binding to mitochondrial complex IV, leading to ETC inhibition and the reduction of mitochondrial energetic production [33,66]. H_2_S catabolism mainly occurs in the mitochondria through oxidation pathways (Figure 2b). From an evolutionary point of view, such mitochondrial location for H_2_S oxidation is unsurprising, owing to the primitive origin of this organelle (a sulfide-oxidizing bacterium). H_2_S clearance begins with its oxidation in the mitochondrial matrix by SQR, producing persulfide (R-SSH), and may occur in all cellular types (Figure 2b) [71]. In physiological steady-state conditions, SQR actively participates in electron transfer to ubiquinone. However, in the oxidation process, H_2_S can donate electrons via SQR, which then follow the traditional transfer route to complexes III and IV and stimulate energetic metabolism [83]. Persulfides are then oxidized by persulfide dioxygenase (ETHE1) to generate sulfites (SO_3_^2−^). Sulfites are further oxidized by sulfite oxidase (SO) to sulfates (SO_4_^2−^), or by rhodanese (Rhod) to thiosulfates (S_2_O_3_^2−^), which can also be reduced to sulfites and H_2_S by thiosulfate reductase (TR) in presence of GSH [84]. The terminal catabolic products of H_2_S—sulfates—are finally eliminated in urine. Additionally, persulfides generated by SQR can also be directly converted to thiosulfates by sulfur transferase (SR) [84]. In physiological conditions, H_2_S oxidation is very fast and efficient, maintaining a low level of H_2_S in tissues (i.e., half-lives of 2.0, 2.8, and 10.0 min in liver, kidney, and brain tissues, respectively) [64]. However, in hypoxic conditions, H_2_S oxidation decreases and H_2_S concentration logically increases, suggesting that H_2_S could be an efficient oxygen sensor [85].

Although considered to be minor, and much slower than oxidation, methylation of H_2_S also allows H_2_S elimination (Figure 2b). In the cytoplasm, ubiquitous S-methyltransferase (TMT) converts H_2_S to methanethiol (CH_4_S) and dimethyl sulfide ((CH_3_)_2_S). The latter is then oxidized by Rhod to sulfates, which are excreted via urine [15].

In addition to enzymatic routes, H_2_S can also be eliminated through expiration. Detection of exhaled H_2_S is possible when large amounts are present, such as during sodium sulfide administration [86]. H_2_S concentration in exhaled air is altered in various hypoxic diseases, such as chronic obstructive pulmonary disease (COPD) or asthma [87]. However, there is no standardized method to quantify this parameter, and very few data regarding H_2_S expiration in healthy population are available [88].

Likewise, intracellular L-cysteine concentration is tightly regulated, especially via oxidative pathways. Briefly, when it is excessive, the enzyme cysteine dioxygenase (CDO) oxidizes L-cysteine to cysteine sulfinate (Figure 2b) [89]. The latter is further converted to sulfites thanks to the activity of cytosolic and mitochondrial CAT and SO, respectively, or to taurine by cysteine sulfinate decarboxylase (CSAD) [89].

### 2.3. H_2_S in the Pulmonary Circulation

Hosoki et al. first characterized CSE as the major source of H_2_S in vascular tissues [25]. CSE is abundantly expressed in bovine [38,39], broiler [40], and rat [27,41,42,43,44] PAs. CSE expression was detected in pulmonary vessels early during post-natal lung development in mice [90]. Discrepancies in the localization of CSE expression were reported in the PA walls. Some studies mentioned that PA media and associated smooth muscle cells are the main or even exclusive sites of CSE expression in pulmonary circulation. Indeed, immunohistochemical staining showed that the presence of CSE proteins was limited to the media of rat peripheral lung vascular tissues [41], and was also found in cultured bovine PASMCs [39]. These findings are in accordance with other reports pointing out the expression of CSE in the smooth muscle cells of systemic vascular walls [22,25,27]. However, localization of CSE expression in the endothelium must be considered, because CSE was also detected in the endothelial layer of PAs and primary PAECs from rats [34,42,91,92,93]. Those discrepancies in CSE localization within the PA walls could be explained by interspecies diversity of the studied models, as well as the pulmonary arterial segments considered (proximal versus distal) [18,45]. As a consequence, a dual role of CSE-associated H_2_S production in the media and intima of PAs should be considered.

Regarding CBS, interestingly, CBS mRNAs and proteins were also detected in rat [42,44], broiler [40], and adult mouse [91] PAs. Like CSE, CBS expression was observed in pulmonary vessels during post-natal lung development in mice [90]. CBS protein quantities in rat aortic and PA tissues were similar under physiological conditions [42]. However, unlike CSE, CBS proteins were clearly detected in the endothelial cells [94]—but not in the smooth muscle cells—of bovine PAs [39]. This endothelial predominance of CBS expression is in accordance with the first report of Zhao et al., showing the absence of CBS mRNA in endothelium-free rat PAs [27]. Although CBS mRNA and protein expression were lower than those of CSE in rat PAs [40], the CBS inhibitor aminooxyacetate (AOA) significantly altered H_2_S production [38]. In contrast, inhibition of CSE provided conflicting reports. shRNA-mediated CSE knockdown in rat PAECs significantly decreased H_2_S levels in supernatants [34], whereas DL-propargylglycine (PAG)—a presumed CSE inhibitor—did not inhibit H_2_S production of bovine PA homogenates in the presence of L-cysteine and pyridoxal-5′-phosphate. However, the selectivity of these commonly used pharmacological inhibitors to assess the functional involvement of CSE and CBS in a vascular context remains questionable [95]. Indeed, the selective CBS inhibitor AOA alters both CBS and CSE activity, and is even more efficient than PAG in reducing CSE activity. Altogether, CSE and CBS appear to be involved in endogenous H_2_S production in the pulmonary vasculature, even though the spatial distribution of H_2_S synthesis within arterial wall layers along the pulmonary vasculature remains to be clarified.

Regarding the final enzymatic source, 3-MST mRNA was observed in rat intrapulmonary arteries [43], and 3-MST proteins were detected in various human endothelial cell lines, including PAECs [18]. Immunohistochemical studies from another report indicate that 3-MST proteins are present in both endothelial and smooth muscle cells of small rat PAs [44] whereas less immunoreactivity was noticed in the media and adventitia compared to the endothelial layers of cow and sea lion PAs [39]. This broad distribution of 3-MST was also found in cultured bovine PAECs and PASMCs [39]. Interestingly, in comparison to aortic tissues, a higher 3-MST mRNA expression was quantified in PAs [42]. Cytosolic and mitochondrial CAT (CAT1 and CAT2, respectively) were notably detected in lung and systemic vascular tissues [22,44,96,97,98]. Du et al. firstly indicated CAT1 and CAT2 mRNA presence in rat PAs [96]. Subsequently, CAT1 and CAT2 proteins were found to be expressed in cultured PAECs, PASMCs, and fibroblasts from rat PAs [34,99,100], suggesting a broad distribution within the pulmonary vascular wall. So far, few studies investigated the physiological relevance of CAT activity in H_2_S synthesis in the pulmonary vasculature. Madden et al. suggested an involvement of CAT activity in the generation of H_2_S through the CAT-3-MST pathway during acute hypoxic treatment of small rat PAs in the presence of cysteine and α-ketoglutarate [45]. However, CAT/3-MST coupling cannot be considered to be a direct H_2_S-producing pathway, because of the requirement of reducing molecules—such as endogenous Trx or DHLA—to produce free forms of H_2_S [22,23,24]. Cytosolic and mitochondrial Trx1 and 2 were detected in endothelial and smooth muscle cells from human PAs [101,102]. However, the link between Trx and CAT-3-MST-associated H_2_S production remains to be elucidated in pulmonary circulation.

Finally, H_2_S biogenesis indirectly depends on the level of its main precursor, L-cysteine, which, in turn, depends on the activity of CDO [78]. CDO was found to be greatly expressed and active in liver and kidney tissues, but also in lung tissues [84,97], and CDO proteins were also detected in the media of rat PAs [97].

## 3. The Physiological Role of H_2_S in Pulmonary Circulation

### 3.1. The Role of H_2_S in Lung and Pulmonary Circulation Development

In recent decades, the role of H_2_S in the modulation of respiratory rhythm and the function of epithelial and mucociliary clearance has been demonstrated [87]. Moreover, H_2_S is also a critical actor in lung development, including pulmonary circulation. The main H_2_S-producing enzymes—CBS and CSE—can be found in airway epithelial and pulmonary vessels from the early stages of post-natal development in mice [90]. Indeed, CBS and CSE are dynamically modulated during the first 10 days of mice’s lives—a pivotal period for lung alveolarization. Firstly, genetic ablation of these two enzymes significantly increases small and medium PA muscularization in neonate mice. Secondly, CBS and CSE are involved in pulmonary vascular growth, since their absence decreases CD31 (endothelial cells marker) expression in newborn mice’s lung homogenates. These results were confirmed by treatments with CBS siRNA or PAG, which induced a reduction in the length of tubes formed by human PAECs. GYY4137, a slow-releasing H_2_S donor, also showed a promoting influence on vascular growth. Altogether, H_2_S and both CBS and CSE activities seem to be important for the modulation of pulmonary vessels’ architecture (Figure 3).

3-MST is also primarily expressed in the smooth muscle cells and weakly in the endothelial cells of pulmonary vessels during the lung development of young mice [103]. Nevertheless, genetic deletion of 3-MST does not trigger any structural alteration, suggesting that, unlike CBS and CSE, its presence is not required for normal lung development [103]. Overall, these findings suggest that H_2_S could be a relevant target to treat pulmonary development diseases, such as bronchopulmonary dysplasia (BPD) and associated PH (BPD-PH), which will be discussed in Section 4.2.

### 3.2. The Role of H_2_S in Oxygen Sensing and Hypoxic Pulmonary Vasocontriction

H_2_S and oxygen have been linked for a long time in the history of the evolution of life on Earth. Primitive bacteria initially used sulfides from hydrothermal vents as a source of energy. The rise of the atmospheric oxygen fraction forced eukaryotic organisms to shift to oxidative metabolism, with oxygen as the final acceptor of mitochondrial ETC [85]. Interestingly, CBS’s molecular structure exhibits a prosthetic heme group, which can interact with oxygen molecules according to oxygen partial pressure [69]. Teng et al. demonstrated the mitochondrial accumulation of CBS induced by ischemia/reperfusion or hypoxia challenges [69]. Under ischemic/hypoxic conditions, the decrease of CBS’s oxygenation status modulates its interactions, resulting in alteration of its degradation by Lon protease, and subsequently in H_2_S production in the mitochondria [69]. In the vascular sphere, hypoxia (10% O_2_) firstly increases 3-MST protein expression, and secondly its mRNA levels, suggesting a dual regulation of 3-MST under low oxygen levels [104]. More importantly, 3-MST-associated H_2_S production was involved in the hypoxia-induced migration of human umbilical endothelial cells [104]. These elements highlight the multifaceted interactions between H_2_S and oxygen with a transcriptional and a putative post-transcriptional regulation of H_2_S-producing enzymes.

In contrast to the hypoxic systemic vasodilation, pulmonary circulation responds to hypoxia by a contraction—a so-called “hypoxic pulmonary vasoconstriction” (HPV)—which is a physiological response to drive the distribution of pulmonary capillary blood flow to areas of the lungs with high oxygen availability, in order to maintain correct hematosis. HPV is characterized by biphasic vasoconstriction constituted firstly by a transient contraction, followed by transient relaxation and then a sustained contraction [105]. The identity of the oxygen sensor(s) involved in this critical physiological mechanism remains controversial. Endothelium-free precapillary vessels do constrict in response to hypoxia, demonstrating that sensor(s), transductor(s) and effector(s), are present in the PA walls [106]. Nevertheless, the role of the endothelium should not be neglected. Indeed, PAECs modulate HPV under conditions of sustained hypoxia [107]. ROS produced in the mitochondria by the oxygen-dependent ETC may be critical for HPV, although two opposing hypotheses (decrease or increase of ROS production in PAs during hypoxia, reviewed in [107]) are under discussion, and the question about the potential role of other cellular actors in HPV remains open.

Olson et al. initially hypothesized that hypoxia decreases H_2_S oxidation by SQR in the mitochondria via the attenuation of oxygen-dependent ETC function and, logically, increases H_2_S concentration [85]. Then, the coupling of H_2_S clearance and ETC function in the mitochondria could define H_2_S as a putative oxygen sensor. In this framework, the elevation of the partial pressure of oxygen promotes H_2_S consumption (i.e., H_2_S oxidation) in bovine lung homogenates and PASMCs, with half maximal consumption at a pO_2_ of 3.2 and 6 mmHg, respectively [39]. Beyond reduced H_2_S oxidation by SQR, hypoxia may inhibit ETHE1 activity, which requires oxygen, to oxidize persulfides to sulfites [85], but this hypothesis has not yet been validated in pulmonary circulation. Interestingly, H_2_S production seems to be modulated by oxygen levels since, in rat lung homogenates and small PAs, H_2_S was produced, in the presence of L-cysteine and α-ketoglutarate, under marked hypoxic conditions, but decreased when the oxygen concentration raised [45]. Progressive increase of H_2_S levels was also observed in bovine PASMCs undergoing 24 h of low oxygen, suggesting that this relationship between H_2_S and oxygen could occur for sustained hypoxic challenges [46]. The elevation of H_2_S production under hypoxia could be explained by the mitochondrial reduction status, where the concentration of endogenous reducing molecules such as DHLA increases, triggering the release of H_2_S in the mitochondria from the catabolic intermediary, thiosulfates [60]. This production is quicker than enzymatic H_2_S production from L-cysteine in response to hypoxia, and could thus be an initial event in oxygen sensing [60]. Hypoxia-induced translocation of CBS and CSE to the mitochondria [69,70] may also participate in increased H_2_S production, but no evidence has been shown at this time in pulmonary circulation. Altogether, these reports confirm the elegant theory of H_2_S and oxygen coupling, and may involve a dual regulation of H_2_S production and clearance balance in the mitochondria according to the partial pressure of oxygen (Figure 3). The link between H_2_S and oxygen is undeniable in pulmonary circulation [39,45,46], and was indicated in other systems, such as cardiomyocytes or carotid bodies [83]. Mechanisms underlying the regulation of H_2_S metabolism by oxygen partial pressure are still unclear, and require further experimentation, especially in pulmonary circulation. 

To study the relationship between H_2_S and HPV, the comparison of the acute effects of H_2_S and hypoxia (pO_2_ < 5 mmHg) on the vascular responses of pulmonary vessels from various species was assessed using H_2_S-releasing sulfide salts—sodium hydrosulfide (NaHS), or sodium sulfide (Na_2_S) (1 mM)—and indicated an intriguing similarity between contractile responses to H_2_S and hypoxia, especially in rat PAs [38]. Surprisingly, H_2_S or hypoxia triggered a biphasic response with a transient contraction (Phase 1—Ph1), followed by a transient relaxation and then a sustained contraction (phase 2—Ph2) in rat PAs. These results are consistent with recent work indicating that high concentrations (> 100 μM) of Na_2_S induce a biphasic contraction of rat PAs, confirming a similar pattern to that observed in HPV [108]. To clarify the role of endogenous H_2_S production in HPV, β-cyanoalanine (BCA)—a potent CSE inhibitor—reduces Ph1 contraction and subsequent relaxation of rat PAs preconstricted with norepinephrine [38]. In addition, PAG, but not AOA, decreased the rise of PA pressure in response to hypoxia in rat perfused lung tissue [45]. Knowing the importance of a reducing environment in H_2_S release by mitochondrial thiosulfates, DTT and DHLA increased the amplitude of HPV in bovine PAs [60]. Interestingly, in contrast to the above results, a report by Prieto-Lloret et al. demonstrated that HPV is not inhibited by PAG when supplemented with physiological concentrations of H_2_S precursors [43]. This is consistent with the absence of the effect of CSE genetic deletion on hypoxia-induced elevation of pressure in murine perfused lung tissue [109]. Another discordant point is that the incubation of PAs with DTT does not potentiate, but rather reduces, the amplitude of HPV [43]. These results put initial reports on the role of H_2_S in HPV into perspective, and suggest that this process is not dependent on H_2_S production in PAs (depending on either the CSE pathway or H_2_S release from thiosulfates in a reducing environment). The observed similarity between PA contraction patterns triggered by either exogenous application of H_2_S or hypoxia does not necessarily entail a physiological relationship (direct or indirect), although it could be explained by shared common mechanisms. Mitochondrial ROS production is a key event in HPV, and activates calcium release from the sarcoplasmic reticulum, and subsequent contraction of PAs [107,110]. In a recent study, Prieto-Lloret et al. assessed the mechanistic issue of biphasic H_2_S-induced contraction [108]. Ph1 and Ph2 contraction and relaxation caused by H_2_S were not affected by L-NAME or endothelium denudation, suggesting that H_2_S (30–1000 μM) acts directly on PASMCs. H_2_S treatment of PASMCs increased ROS production, whereas SQR genetic deletion abolished this effect. The sustained Ph2 contraction was inhibited by myxothiazol (a complex III inhibitor), but not by rotenone (a complex I inhibitor). This report thus suggests that H_2_S stimulates the ETC through H_2_S oxidation by SQR, in turn stimulating ROS production from complex III and, thus, triggering the sustained Ph2 contraction. In contrast, this Ph2 sustained contraction induced by hypoxia during HPV is inhibited by rotenone [110], thus suggesting a different mechanism from that triggered by H_2_S. One hypothesis is that, when the intracellular H_2_S level increases, its poisoning effect on complex IV inhibits the ETC and decreases ROS production, thus inducing a transient relaxation. H_2_S evaporation decreases its level and again promotes the ETC via SQR oxidation, increasing ROS production by complex III (sustained Ph2 contraction). Overall, H_2_S’s influence on PA contraction may be due to a balance between its promoting and blocking effects on ETC function (Figure 3).

Initial reports indicated opposite dose-dependent effects of H_2_S on preconstricted bovine PAs [38]. Between 10 nM and 10 μM, H_2_S relaxes, whereas above 10 μM of H_2_S induces a constriction of bovine PAs. However, it is noteworthy that in most tissues and in plasma steady-state H_2_S levels do not exceed the nanomolar range [63,64]. In this range, H_2_S preferentially relaxes PAs. A putative increase in intracellular H_2_S levels from nanomolar range to 10 μM in response to hypoxia would induce detrimental consequences on the mitochondria. Numerous considerations about variability in the experimental approaches (H_2_S-releasing molecules, concentration, agonist-induced pre-tone of the vessel, oxygen levels, animal species, etc.) must be considered in order to explain opposing reports, and the development of reliable methods to control and measure intracellular H_2_S levels are required. As previously mentioned, the use of pharmacological inhibitors of H2S enzymatic production also represents a significant issue, due to their relative specificity and the detrimental absence of selective CBS inhibitors [95].

### 3.3. The Role of H_2_S in Pulmonary Artery Relaxation

Numerous studies associate H_2_S with the regulation of vascular tone. The relaxing effects of exogenous H_2_S in portal veins and thoracic aortae of rats [25,27] were initially reported, and were associated with decreased arterial blood pressure [27,28,109]. For instance, CSE genetic deletion in mice reduces endothelium-dependent vasorelaxation, suggesting an important role of endogenous H_2_S production in this process [28].

Initial reports of the effect of exogenous H_2_S on PAs’ vascular tone indicated that, at physiological levels, H_2_S relaxes bovine PAs in a dose-dependent manner, with a half maximal effective concentration of 550 ± 180 nM [38] (Figure 3). High concentrations of H_2_S (300 μM) also relax bovine PAs preconstricted by severe hypoxia [38]. NaHS induces a dose-dependent relaxation in rat PAs [42,47]. Exogenous H_2_S (20–500 μM)’s relaxing effects were also observed in isolated large- and medium-sized PAs from human donors, and were associated with a decrease in PA pressure in human perfused lung tissues [111]. However, it should be noted that the genetic deletion of CSE in mice does not impact steady-state mPAP in isolated perfused lung tissues [109], suggesting that endogenous H_2_S production by CSE is not involved in the regulation of basal mPAP, at least in mice. Derived organosulfur compounds from garlic, such as allicin, diallyl disulfide, and diallyl trisulfide, have for a long time been associated with H_2_S release under physiological conditions, triggering vascular relaxation in systemic vessels and decreasing blood pressure [58,112]. In pulmonary vascular beds, allicin—but not diallyl disulfide or trisulfide—dose-dependently relaxed rat PAs and reduced PA pressure in rat perfused lung tissues [113,114]. Interestingly, pre-treatment with garlic extracts significantly decreased endothelin-1 contraction of rat PAs [115]. However, the contribution of H_2_S release from these garlic-derived compounds to PA relaxation remains to be elucidated.

Cellular mechanisms underlying the relaxing effect of H_2_S on PAs were primarily associated with K_ATP_ channel activity. Indeed, Zhao et al. demonstrated that H_2_S promotes opening of K_ATP_ channels, leading to smooth muscle cell hyperpolarization and, consequently, aortic relaxation [27]. This result was then reproduced in pulmonary circulation, where glibenclamide (a blocker of plasma membrane K_ATP_)—but not 5-hydroxydecanoate (blocker of mitochondrial K_ATP_)—partially inhibited H_2_S-induced PA relaxation in rats [42]. H_2_S has been shown to directly interact with the Kir6.1 regulating subunit of K_ATP_ channels through S-sulfhydration (post-translational modification consisting in the formation of a persulfide group (R-SSH) on a cysteine residue) of cysteine residue (cys43) [116]. This modification of the K_ATP_ regulating subunit alters ATP binding and, consequently, K_ATP_ activity is enhanced, promoting hyperpolarization and vascular relaxation [116]. Although Kir6.1 was found in rat PAECs and PASMCs [42], no direct evidence links Kir6.1 S-sulfhydration and H_2_S-induced PA relaxation. In aortic tissue, H_2_S inhibits mitochondrial metabolism, thus resulting in decreased ATP production [117]. Alteration of ATP levels promoting K_ATP_ activity could be another mechanism in H_2_S-induced PA relaxation. H_2_S has long been known to interact with other gasotransmitters, such as NO, through chemical or metabolic interplays [118]. For instance, H_2_S-induced relaxation of rat PAs is altered by Nω-nitro-L-arginine methyl ester (L-NAME), a potent NO synthase inhibitor [47]. PA relaxation triggered by sodium nitroprusside—an NO donor—is also partially reduced by PAG. Allicin-mediated PA relaxation is also dependent on the NO pathway [115]. Endogenous NO production thus seems to be involved in H_2_S relaxation, and vice versa, establishing a possible metabolic crosstalk for PA tone regulation (Figure 3). Although precise mechanism(s) remain to be determined in the pulmonary vascular bed, these results are in accordance with numerous reports showing H_2_S and NO pathway interactions in the modulation of systemic vascular tone [25,28,119,120]. Owing to the significant relaxing influence of H_2_S, many studies have shown a protective effect of H_2_S on increased mPAP during PH development, which will be discussed in the following section.

## 4. The Role of H_2_S in Hallmarks of PH 

Alterations in H_2_S production or clearance lead to a pathological drift contributing to vascular injury, notably through endothelial dysfunction, inflammation, oxidative stress, decreased vasorelaxation, vascular remodeling, and platelet aggregation (already reviewed in [29,32,121,122]). Depending on the considered pathological condition, circulating H_2_S levels are either decreased or increased in patients [123,124,125]. High levels of homocysteine that promote endothelial dysfunction are also observed in various clinical cardiovascular conditions [122]. Therefore, the global vasculoprotective role of exogenous H_2_S supplementation—especially through its anti-inflammatory, antioxidative, and vasorelaxant properties—argues in favor of H_2_S acting as a new therapeutic strategy for vascular diseases. In view of the implications of H_2_S in pulmonary circulation, we will discuss H_2_S metabolism alterations in the setup of PH, as well as the potential protective role of H_2_S in the development of associated pathological hallmarks.

### 4.1. H_2_S Metabolism Alterations in PH

#### 4.1.1. H_2_S Metabolism in Human PH

According to PH classification (5 groups), PA pathological changes can be associated with a multitude of clinical conditions [2]. Congenital heart diseases (CHDs), such as Eisenmenger syndrome or systemic-to-pulmonary shunts, can promote PH development due to increased blood flow within pulmonary circulation (PH group 1) [2]. In children with CHD-associated PH, circulating H_2_S levels and CSE expression are lower than in children without PH [126]. In contrast, homocysteine levels are increased, but the low H_2_S levels may be due to the reduced CSE expression, thus inducing a decrease in homocysteine conversion into L-cysteine via the transsulfuration pathway [126,127]. In addition, low endogenous H_2_S was correlated with worse prognosis after surgical correction of CHD [128]. These results suggest that H_2_S and actors of its metabolism (namely homocysteine and CSE) could be potential biomarkers to determine the short-term prognosis and risk of CHD complicated with PH.

Chronic hypoxia exposure triggers a cascade of pathological manifestations inducing PH, characterized by a sustained PA contraction and decreased relaxation (endothelial dysfunction), PA remodeling, and inflammation, finally leading to right ventricle hypertrophy [4]. Indeed, lung hypoxic chronic diseases such as COPD, sleep apnea, or fibrosis/emphysema disorders are associated with PH development (PH group 3) [2,4]. PH is commonly observed in severe COPD, seen in more than 90% of patients with mPAP > 20 mmHg [129]. COPD is mainly characterized by a marked inflammation of the airway tract, lung parenchyma, and pulmonary vasculature, contributing to exacerbations and leading to possible pathological drift towards PH. An initial report mentioned that H_2_S serum levels in patients with acute COPD exacerbation are lower than those with stable COPD [130]. In addition, H_2_S levels are progressively reduced with the decrease of airway obstruction severity, and when systolic PAP is greater than or equal to 35 mmHg in COPD patients with acute exacerbation [130]. In accordance with these results, a complementary study indicated that CSE protein and CBS mRNA expression were significantly decreased in COPD patients [131]. A reduction in exhaled H_2_S levels was also found in COPD patients with significant quantities of eosinophils, suggesting a potential link with the modulation of inflammation [132].

BPD is a common lung developmental complication in premature newborns, characterized by an arrest of alveolarization and a decrease in angiogenesis due to the requirement of mechanical ventilation and hyperoxic treatment [133]. Those developmental alterations can lead to decreased pulmonary vascular density and promote PA muscularization and inflammation [134,135]. As a consequence, increased PA resistance and pressure can be observed, and 25–40% of premature infants with severe BPD will develop PH (BPD-PH). In case of BPD-PH, the death rate after 2 years is between 33 and 48% [136]. Although there are multiple pathophysiological mechanisms, dysregulation of H_2_S metabolism was demonstrated in the context of BPD. Indeed, cystathionine plasma levels are higher in premature than in full-term newborns. In contrast, L-cysteine levels are significantly lower in premature infants [137]. Moreover, hepatic CSE activity is lower in premature than in mature infants [138]. This result brings evidence of a possible alteration of the transsulfuration pathway in pre-term newborns. However, the link between dysfunction of H_2_S metabolism and the risk of developing BPD-associated PH is not yet elucidated.

Altogether, these results highlight an alteration of the endogenous H_2_S metabolism in CHD, COPD, and BPD, and could constitute an interesting biomarker of severity, prognosis, and risk to develop increased PAP.

#### 4.1.2. H_2_S Metabolism in Experimental PH Models

To understand the complex pathophysiology of PH, various in vivo experimental models were developed to mimic clinical PH representations. Each experimental model has its own pathophysiology and etiology, although all exhibit, with different intensity, the primary markers of PH—namely, pronounced PA remodeling, increased mPAP, inflammation, and RV hypertrophy [139]. The monocrotaline (MCT) animal model consisted of one injection of MCT—a pyrrolizidine alkaloid extracted from the plant *Crotalaria spectabilis*—metabolized into its active form (MCT pyrrole) in the liver by monooxygenase. MCT triggers pulmonary vascular injury, mainly via endothelial dysfunction and exacerbated inflammation, leading to increased mPAP and both PA and RV remodeling [140]. Interestingly, MCT-treated rats exhibit lower H_2_S levels in plasma and in lung tissue [34,141]. CSE protein expression and activity in lung tissue are also decreased in comparison to control rats [50,141]. These alterations are time-dependently downregulated in this PH model, with significant reductions in plasma H_2_S levels, lung CSE expression, and activity from 14 days after MCT injection [50]. A treatment of human PAECs with the MCT pyrrole decreases H_2_S production and CSE protein expression [141], suggesting a possible alteration of the endogenous H_2_S metabolism in PA endothelium.

In order to model secondary PH due to chronic hypoxic diseases such as COPD, animals are subjected to hypoxia housing in normobaric or hypobaric chambers. Similarly to the MCT PH model, daily exposure to normobaric or hypobaric hypoxia (10% O_2_) for 3 weeks was associated with a significant decrease of H_2_S plasmatic concentration and lung H_2_S production [35,40,53,142,143]. Moreover, expression of CSE mRNA was altered in lung tissues [35,40]. Interestingly, López et al. compared the effects of altitude exposition (hypobaric hypoxia) on newborn sheep versus newborn llamas, which are adapted to chronic exposition to hypoxia at high altitudes [144]. First of all, altitude exposition increased homocysteine plasma levels in sheep, but not in llamas, which presented basal low levels of homocysteine regardless of altitude. In addition, increased PA pressure and resistance were observed in sheep, but not in llamas, in response to altitude. Thus, basal low levels of homocysteine in llamas could be explained by their catabolism by CSE and CBS via the transsulfuration pathway, and may prevent the effects of hypoxic exposure and associated PH. On the other hand, disruption of this route by an undefined hypoxia-dependent mechanism could trigger homocysteine accumulation and attenuate H_2_S bioavailability, promoting vascular injury and PH setup. PH secondary to COPD and emphysema has also been studied in mice subjected to tobacco smoke [94]. After 12 or 24 weeks of tobacco smoke exposure, H_2_S concentration was unchanged, whereas the capacity to produce H_2_S was decreased in murine lungs [94]. Such results could be explained by the significant reduction of CSE and CBS protein expression [94].

Beyond commonly used PH models, experimental PH can also be provoked by creating a shunt between the abdominal aorta and the inferior vena cava in order to increase pulmonary blood flow. After 11 weeks, rats with high pulmonary blood flow exhibited reduced plasma and lung H_2_S levels, as well as lung H_2_S production rates [92,145]. CSE, CBS, and 3-MST protein expression were also reduced in the PAs of these rats [44]. Surprisingly, four weeks after shunt, H_2_S concentration in the lung tissue was higher than in control rats. This elevation was suppressed with daily treatment with PAG (a CSE inhibitor), suggesting an increase in H_2_S production caused by CSE [93,145]. Indeed, we can hypothesize that the increase in H_2_S content in the lung tissue after four weeks could act as a compensatory mechanism to counteract the high PA pressures, perhaps via the relaxant properties of H_2_S on PAs. Prolongation of high PAP could induce a decompensation state, with a decrease in H_2_S production through the alteration of CSE, CBS, and 3-MST expression. Such hypotheses imply a cellular relationship between blood pressure detection and H_2_S metabolism. Already mentioned in recent reports on systemic vessels [81,146], this association needs to be demonstrated in pulmonary circulation.

Overall, experimental models of PH tend to reproduce clinical observations of dysregulated H_2_S metabolism in PH, whatever the PH group considered. Circulating and pulmonary H_2_S contents are decreased in most reports, and associated with altered expression and/or activity of H_2_S-generating enzymes. Dysregulation of CSE seems to be a pivotal common factor, since it is associated with various forms of PH in human and experimental models alike. Considering the major role of CSE in vascular tissues, and its involvement in endogenous H_2_S synthesis in PAs (see Section 2.2), reduction of its expression and/or activity in the lungs could be associated with high homocysteine levels and decrease of H_2_S vascular bioavailability. Finally, under pathological conditions, H_2_S metabolism alterations could, at least in part, result in PA injury through the development of endothelial dysfunction, remodeling, and decreased vasorelaxation. Altogether, current clinical and experimental research converges to a critical protective role of H_2_S metabolism in the pathophysiology of PH, which will be discussed in next section.

### 4.2. H_2_S Exerts Protective Effects against PH

Research into the therapeutic potential of H_2_S against PH is based on H_2_S supplementation experiments using H_2_S-releasing molecules. The most commonly used are the sulfide salts NaHS and Na_2_S, which are inexpensive, water-soluble, and quickly release large amounts of H_2_S under physiological conditions [54,55]. Slow-releasing H_2_S donors, such as GYY4137 (morpholin-4-ium 4-methoxyphenyl (morpholino) phosphinodithioate), were further developed, and already exhibit interesting effects on cardiovascular diseases [26,54,55]. The beneficial effects of H_2_S-releasing molecules’ administration on experimental PH development are summarized in Table 1.

In the MCT model, preventive daily injection of NaHS (56 μmol/kg, intraperitoneal) for 21 days significantly reversed the reduced H_2_S levels in plasma and in lung tissues, and decreased mPAP, RV hypertrophy, and PA remodeling via the reduction of media thickness [34,141]. This protective effect was also reported with a curative treatment with NaHS (1 mg/kg, intraperitoneal) 7 days after MCT injection [50]. More surprisingly, decreased CSE protein expression was also reversed in rat lung tissue, suggesting a feedback influence of H_2_S on its production routes [141]. In the same manner, low CSE protein expression observed in human PAECs treated with MCT pyrrole in vitro was reversed by the addition of H_2_S [141]. The protective influence of NaHS was associated with an inhibition of PH-associated inflammation, as shown by reduction of the plasmatic and pulmonary contents of the proinflammatory cytokines TNF-α, IL-6, and IL-8 [34,141]. H_2_S supplementation attenuated the activation of the nuclear factor kappa B (NF-κB) pathway—a pivotal signaling pathway in inflammation in PH [147,148]—in lung tissue from rats with MCT-associated PH [141] and in PAECs treated with MCT pyrrole [37,141]. Moreover, NaHS treatment reduces α-smooth muscle actin and increases VE-cadherin expression in the PAs of rats injected with MCT [50]. This effect was aggravated when rats were treated with PAG. NaHS also dose-dependently inhibited the in vitro phenotypic shift of human PAECs into mesenchymal cells induced by TGF-β1 treatment [50]. This process was mimicked by CSE overexpression. Altogether, these results suggest that H_2_S inhibits the endothelial–mesenchymal transition and associated remodeling observed in PH. In contrast to NaHS, dithiolthione (ADT-OH) was demonstrated to slowly release H_2_S in vivo [149]. Interestingly, 7 days after MCT injection, daily inhalation of ACS14 (a conjunction of ADT-OH and aspirin) encapsulated in a large porous microsphere decreased mPAP, PA remodeling, and RV hypertrophy, similarly to sildenafil, which is known to decrease such PH hallmarks [56]. Like NaHS, ACS14 treatment also reduced endothelial–mesenchymal transition in the PA walls of rats injected with MCT.

In the hypoxia model, daily administration of NaHS (14 μmol/kg) also demonstrated promising properties in PH. NaHS markedly inhibited mPAP in rats and broilers with hypoxic PH [35,40,53,142,143]. PA remodeling was also improved, with reduction of media thickness, the number of muscularized PAs, and the presence of collagen types I/III and elastin in PAs [35,143]. In addition, NaHS increased the total antioxidant capacity of lung homogenates, highlighting an improvement in cellular defenses against hypoxia-induced oxidative stress [53]. As in MCT models, NaHS treatment succeeded in reversing altered plasma and pulmonary H_2_S levels [35,40,53,142,143]. In the murine tobacco smoke-induced PH model, daily treatment with NaHS (50 μmol/kg) decreased TNF-α amounts in bronchial alveolar lavage- and 8-hydroxyguanine (a marker of DNA injury induced by ROS)-positive cells, suggesting an attenuation of lung inflammation and oxidative stress. Moreover, NaHS treatment restored murine lung expression of CSE and CBS proteins due to Akt protein activation [94]. Interestingly, these results were associated with a reduction in RV systolic pressure [94]. In another model of rats with COPD obtained via smoke exposure and lipopolysaccharide tracheal instillation, Ding et al. showed the beneficial effects of NaHS (56 μmol/kg) on PAEC apoptosis and associated endothelial injury [150]. Furthermore, a recent work analyzed the therapeutic potential of a preventive treatment with a slow-releasing H_2_S donor—GYY4137—on hypoxic PH [51]. Partial reduction of mPAP and total PA resistance (mPAP/cardiac output ratio) was reported in rats subjected to 4 weeks of hypoxia and treated with GYY4137. PA media thickness was also decreased, which is consistent with the inhibition by GYY4137 (100 μM) of hypoxia-induced PASMC proliferation and migration, without inducing apoptosis [51]. ER stress was recently considered to be a pivotal manifestation during PH setup, and notably characterized by stimulation of the activating transcription factor 6 (ATF6) and disruption of ER–mitochondria interactions [151]. GYY4137 inhibited the expression of ER-stress-associated proteins binding the immunoglobulin protein (Grp78) and ATF6 in PAs [51]. However, as already demonstrated in HUVEC [152], the influence of GYY4137 on the ER–mitochondria unit remains to be defined in PA walls. It is important to note that, in addition to vascular benefits, NaHS and GYY4137 also demonstrated protective effects on cardiac function and remodeling, both in hypoxic PH and in PH secondary to COPD [35,51,53,94]. Beyond NaHS and GYY4137, garlic was studied for its short-term hemodynamic properties on pulmonary circulation. Indeed, daily treatment of rats with garlic by gavage (100 mg/kg) for 5 days significantly reduced the increase in mPAP induced by 90 min of hypoxia housing (10% O_2_), without modifying systemic arterial pressure [57]. This effect was related to the relaxant influence of garlic on PAs (see Section 3.3).

An experimental model of BPD-PH was induced by exposing mouse or rat pups to hyperoxia during their first days of life. Such conditions promote lung inflammation, alveolar/vascular growth inhibition, increase in PA wall thickness and, consequently, PH [52,153]. Vadivel et al. assessed the benefits of daily administration with GYY4137 on hyperoxia (95% O_2_)-induced BPD-PH in rat pups [52]. This treatment reduced PASMC proliferation, PA media remodeling, and RV afterload, and partially attenuated RV hypertrophy, suggesting that GYY4137 could prevent vascular and cardiac adverse manifestations in BPD-PH. This beneficial effect was associated with an improvement of alveolar growth and pulmonary vessel density in lungs. Vascular growth alteration under hyperoxia is primarily due to a reduced networking of PAECs. Interestingly, GYY4137 treatment improved human PAEC network formation in normoxia or hyperoxia [52,90]. In relation to these results, GYY4137 also improved cell viability and reduced oxidative stress induced by hyperoxia in human PAECs, suggesting that such a compound could have a beneficial effect on PA remodeling [52]. The protective effects of H_2_S against BPD-PH are in total accordance with H_2_S’s significant role in both lung and pulmonary circulation in post-natal development (see Section 3.1).

Finally, in the model of high pulmonary blood flow PH, H_2_S donors also revealed interesting properties. Indeed, daily NaHS (56 μmol/kg) administration during the 11 weeks of shunting partially attenuated increased mPAP and RV hypertrophy [48,143,145]. Improvement of PA structural alteration was observed with a reduction of PA remodeling and collagen type I/III staining in the PA walls [48]. In PA media from rats treated with NaHS, an increase in apoptosis markers—such as Fas and caspase-3—was observed [145]. It must be noted that NaHS also reversed the increased lung endothelin-1 levels—a potent vasoactive and pro-proliferative agent [48].

Beyond vascular manifestations, PH is also characterized by critical structural alterations (e.g., inflammation, fibrosis, remodeling, etc.) of the RV, resulting in RV failure and death [3]. H_2_S administration in PH experimental models showed beneficial effects on RV hypertrophy (see Section 4.2). Nevertheless, mechanisms underlying this influence are still undetermined; two options may merit further examination—namely, a direct effect on the myocardium, and/or an indirect process via attenuation of the RV afterload. Numerous studies reported the relationship between decreased plasma H_2_S levels and myocardial infarction or heart failure, suggesting an intrinsic role of H_2_S metabolism in cardiac homeostasis [154]. In both experimental and clinical cases, H_2_S administration using SG-1002—a slow-releasing H_2_S donor—exerted cardioprotective effects on heart failure through a proposed increase of both circulating NO bioavailability and the eNOS pathway [155,156]. In addition, liposomal ZYZ-802—another slow-releasing H_2_S molecule—reduced collagen fiber amounts and associated fibrosis in the myocardium in a heart failure model of rats [157].

At the mitochondrial level, PH is characterized by a cancer-like metabolic shift (Warburg effect) from oxidative phosphorylation to glycolysis in both PA and RV cells. Mitochondrial fission through increased dynamin-related protein 1 (DRP1) and decreased mitofusin 2 (MFN2) has also been demonstrated [158]. Interestingly, H_2_S oxidation by SQR stimulates oxidative phosphorylation, presenting the mitochondria as a privileged target of H_2_S [65,80]. H_2_S can also interact with the expression and function of proteins regulating mitochondrial dynamics. In fact, H_2_S decreases DRP1 and enhances MFN2 expression in the myocardium, and subsequently improves mitochondrial ultrastructure and function [159,160]. These beneficial effects were associated with an attenuation of myocardial hypertrophy in mice [159,160]. Moreover, CBS knockdown or inhibition using AOA was associated with a reduction of MFN2 expression in ovarian cancer cells [161]. In relation to its role in oxygen sensing and hypoxia responses in PAs (see Section 3.2), H_2_S could thus be of interest to reduce or reverse mitochondrial alterations associated with PH.

In summary, H_2_S-releasing molecules show significant protective effects on vascular and cardiac manifestations of various forms of PH in experimental PH models. H_2_S’s beneficial effects on PH are multifaceted on various hallmarks, such as inflammation, PA remodeling (endothelial–mesenchymal transition; migration and proliferation of cells from the PA walls), and oxidative and ER stress (Figure 4). H_2_S-based preventive and curative treatments improved endogenous H_2_S production, especially by CSE, which appeared to be critical in the regulation of PH hallmarks. To date, the lack of studies using GYY4137 or other slow-releasing H_2_S donors on PH does not allow for comparison of their effects with those of sulfide salts, such as NaHS. However, slow-releasing H_2_S donors seem to better control H_2_S rates, thus avoiding high H_2_S levels, which could be deleterious to mitochondria [54,55]. Although NaHS decreases lung ET-1 levels [48], the effect of preventive H_2_S donor supplementation on the decreased PA relaxation and hyper-reactivity observed in PH remains to be assessed.

## 5. Conclusions and Clinical Perspectives

### 5.1. H_2_S Significance and Perspectives in PH

Like NO and CO, the effect of H_2_S in the regulation of vascular homeostasis and fundamental functions is undeniable. H_2_S production takes part in pulmonary circulation development and oxygen sensing, and acts as a potent vasodilator. Alterations of endogenous H_2_S metabolism are observed in various clinical cardiovascular conditions [123,124,125], including PH and associated diseases such as CHD, COPD, and BPD (see Section 4.1.1). Although the cellular mechanisms of these dysregulations remain unclear, decreased H_2_S bioavailability is critical in the promotion of endothelial dysfunction, exacerbated inflammation, oxidative stress, and changes in the proliferative behavior of smooth muscle cells in the PA walls. In future clinical investigations, H_2_S and associated metabolism actors (namely, L-cysteine, homocysteine, CSE, CBS, and 3-MST) could thus be considered to be relevant biomarkers to characterize prognosis and risk of developing PH from CHD, COPD, or BPD. Enzymatic pathway production is considered to be the major route of H_2_S production in PAs. However, non-enzymatic H_2_S production by red blood cells from L-cysteine and iron has been recently evidenced as a new source of circulating H_2_S under physiological conditions [59]. Interestingly, patients with severe PH exhibit iron deficiency [162]. Hypothetically, reduced iron levels could lead to a reduction of non-enzymatic H_2_S production by red blood cells and, consequently, decreased circulating H_2_S bioavailability in vascular tissues. Although interesting, this hypothesis needs to be experimentally proven in the context of PH.

### 5.2. H_2_S-Releasing Molecules as a New Therapeutic Strategy for PH?

As discussed above, endogenous H_2_S alterations in PH patients and the beneficial impacts of H_2_S on the experimental pathophysiology of PH raise the issue of its clinical potential interest for PH. In the past few years, multiple pharmacological tools have been developed in order to manipulate H_2_S under physiological conditions. Sulfide salts were shown to release H_2_S in large amounts, allowing for quick and efficient distribution. In healthy humans, Toombs et al. showed that intravascular administration of Na_2_S induces an increase in the blood concentration of H_2_S and thiosulfates, as well as exhaled H_2_S levels, during the first minutes after injection [86]. However, this H_2_S bolus released by sulfide salts could activate the detrimental effects of H_2_S on mitochondrial function if high concentrations are reached, especially with susceptible individuals, as in patients with PH. Slow-releasing H_2_S donors, such as GYY4137 or dithiolthione compounds, thus represent an attractive alternative, since they exhibit interesting effects with prolonged liberation in vivo [55,149]. This characteristic avoids the bolus effect of sulfide salts, and allows a controlled release and a greater bioavailability of H_2_S to mimic endogenous H_2_S production rates and maintain a relevant concentration for a long period of time. Interestingly, numerous H_2_S donors are currently considered to be safe for a clinical application. Natural H_2_S donors, such as garlic extracts, have already demonstrated significant influence in lowering the blood pressure of hypertensive patients in multiple clinical trials [163]. Although efficient and highly safe, garlic supplementation should be considered as more of a co-treatment than a unique strategy, because of the lack of hindsight on long-term cardiovascular efficiency [163]. Anethole trithione (CAS number 532-11-6)—a dithiolthione compound—has been widely used and marketed for decades to treat salivary deficiency, cholecystitis, and hepatitis, with no known major side effects [164]. The structural versatility of dithiolthione compounds offers a multitude of plausible strategies to conjugate their H_2_S-releasing properties with other active molecules to improve PH hallmarks [54]. In this framework, dithiolthione was conjugated with sildenafil (ACS-6, http://www.ctgpharma.com, accessed on 10 June 2021) to associate the vasculoprotective features of H_2_S with an established treatment of PH (sildenafil) [165]. As sildenafil, ACS-6 inhibits phosphodiesterase type 5 (PDE5) activity and TNF-α-induced superoxide formation in PAECs in vitro [165]. H_2_S release by ACS-6 is more sustained than that by NaHS. Nevertheless, the preclinical potential of this molecule on experimental PH models has not yet been assessed. Other H_2_S-liberating molecules have been, or are being, studied in various cardiovascular clinical contexts, and subjected to patenting [166]. For instance, the aforementioned SG-1002 (clinicaltrials.gov; ID: NCT01989208) has been clinically tested to counteract H_2_S deficiency in heart failure patients. This compound is well tolerated at various doses, and no changes on hemodynamic parameters or clinical chemistry have been found in healthy or heart failure patients [155].

In summary, H_2_S-releasing molecules have already shown promising features to attenuate vascular and heart alterations in PH. Nevertheless, H_2_S donor research is emerging, and clinical data in the context of PH are still lacking. It will be pertinent to compare the efficiency of various H_2_S-releasing molecules on the development of PH hallmarks in experimental models. Moreover, it will be relevant to decipher the specific actions and mechanisms of such treatment on RV functional and structural alterations during PH development. Innovative synthetic as well as natural H_2_S-releasing molecules, with their broad spectrum of action on vascular and cardiac manifestations, thus constitute a relevant opportunity to develop new therapeutic strategies for PH, and further preclinical investigations on animal models are required prior to any clinical implementation.

## Figures and Tables

**Figure 1 cells-10-01477-f001:**
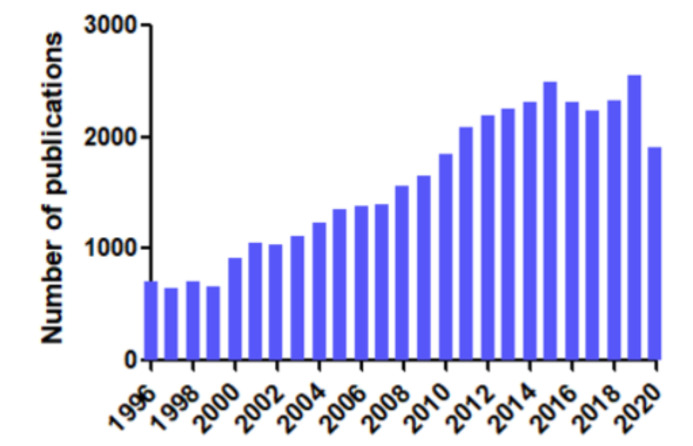
Rising number of publications in H_2_S research fields. A search for original and review articles published from 1996 to 2020 was performed in the PubMed database (www.ncbi.nlm.nih.gov/pubmed, accessed on 10 June 2021) using the key words “hydrogen sulfide”.

**Figure 2 cells-10-01477-f002:**
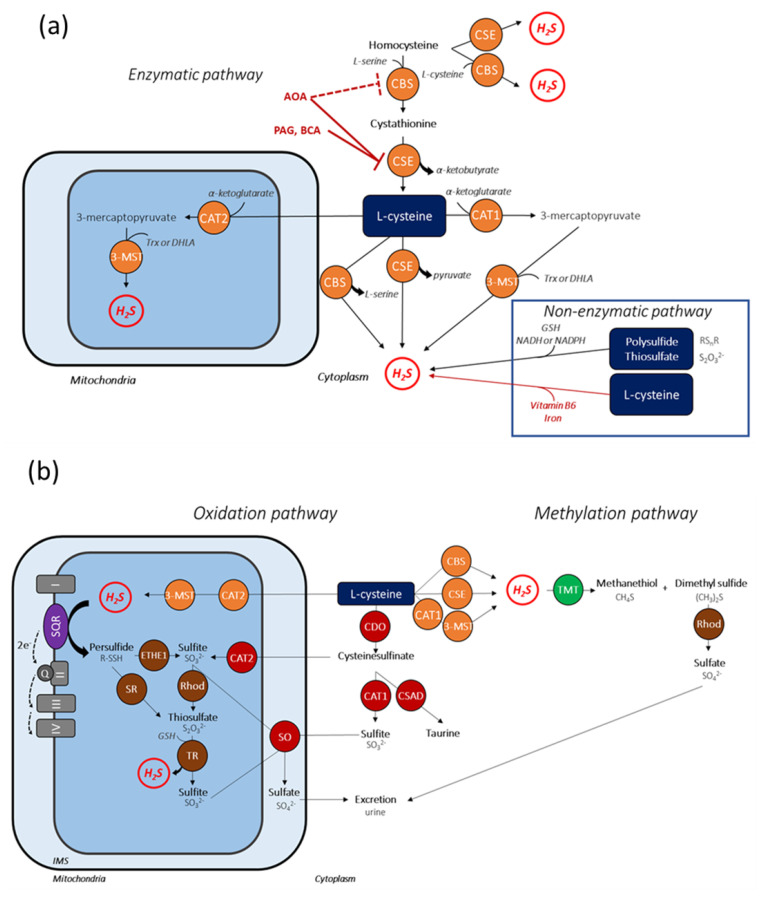
Main H_2_S anabolic and catabolic pathways in mammalian cells. (**a**) Endogenous hydrogen sulfide (H_2_S) production is mainly due to enzymatic pathways through the activity of cystathionine γ-lyase (CSE), cystathionine β-synthase (CBS), and 3-mercaptopyruvate sulfur transferase/cysteine aminotransferase (3-MST/CAT) coupling. Cytosolic CBS and CSE are involved in the interconversion of homocysteine to L-cysteine (transsulfuration pathway). In the context of cysteine catabolism, CBS and CSE desulfhydrate L-cysteine (and homocysteine) to produce H_2_S. L-cysteine can also act as a substrate with α-ketoglutarate to produce 3-mercaptopyruvate through a transamination reaction via activity of the CAT1 and CAT2 enzymes (cytosolic and mitochondrial, respectively). CSE activity is selectively inhibited by DL-propargylglycine (PAG) and β-cyanoalanine (BCA). CBS and CSE activities—especially the latter—are both inhibited by AOA. In a reducing environment (presence of thioredoxin (Trx) and dihydrolipoic acid (DHLA)—two endogenous reducing molecules), 3-mercaptopyruvate is then used by 3-MST to release pyruvate and H_2_S, mainly in the mitochondria. To a lesser extent, non-enzymatic H_2_S production can occur in physiological conditions, caused by reactive sulfur groups of thiosulfates (S_2_O_3_^2−^) or polysulfides (RS_n_S) in the presence of glutathione (GSH) or reducing equivalents (nicotinamide adenine dinucleotide (NADH) and nicotinamide adenine dinucleotide phosphate (NADPH)). (**b**) Oxidation, the major catabolic pathway to maintaining H_2_S homeostasis, occurs in the mitochondria. H_2_S is rapidly oxidized by sulfide quinone oxidoreductase (SQR) to form persulfides (R-SSH), which undergo another oxidation step by persulfide dioxygenase (ETHE1) to produce sulfites (SO_3_^2−^). In this process, two electrons (e^−^) are released to ubiquinone (Q) and transferred to complex III of the mitochondrial electron transfer chain (ETC) [33,83]. Sulfites are either converted to thiosulfates or directly to sulfates (SO_4_^2−^), thanks to rhodanese (Rhod) and sulfite oxidase (SO), respectively. The final catabolic products—sulfates—are finally excreted via urine. In an additional pathway, persulfides can be degraded to thiosulfates by sulfur transferase (SR). Thiosulfates can also be converted to sulfites and H_2_S by thiosulfate reductase (TR) in the presence of GSH, which, ultimately, leads to sulfate production by SO. H_2_S methylation is the other, although minor, clearance pathway. H_2_S is first converted to methanethiol (CH_4_S) and dimethyl sulfide ((CH_3_)_2_S) via S-methyltransferase (TMT) activity. Then, Rhod breaks down dimethyl sulfide into sulfates, which are then excreted through urine. L-cysteine is degraded to cysteine sulfinate by cysteine dioxygenase (CDO) activity. Cysteine sulfinate is then converted to sulfites by CAT1/2 and, ultimately, to sulfate by SO, or to taurine by cysteine sulfinate decarboxylase (CSAD). 3-MST: 3-mercaptopyruvate sulfur transferase; AOA: aminooxyacetate; BCA: β-cyanoalanine; CAT1/2: cysteine aminotransferase 1/2; CBS: cystathionine β-synthase; CDO: cysteine dioxygenase; CSE: cystathionine γ-lyase; CSAD: cysteine sulfinate decarboxylase; DHLA: dihydrolipoic acid; e^−^: electron; ETC: electron transfer chain; I, II, III, and IV: mitochondria complexes I–IV of ETC; ETHE1: persulfide dioxygenase; GSH: glutathione; H_2_S: hydrogen sulfide; IMS: intermembrane space; NADH: nicotinamide adenine dinucleotide; NADPH: nicotinamide adenine dinucleotide phosphate; PAG: DL-propargylglycine; Q: ubiquinone; Rhod: rhodanese; SO: sulfite oxidase; SQR: sulfide quinone oxidoreductase; SR: sulfur transferase; TMT: S-methyltransferase; TR: thiosulfate reductase; Trx: thioredoxin.

**Figure 3 cells-10-01477-f003:**
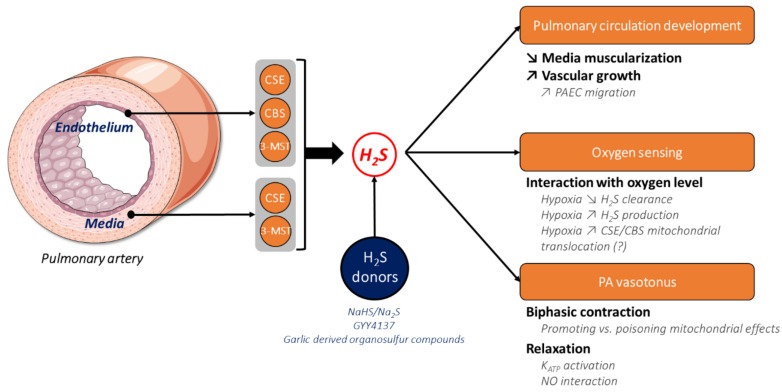
Endogenous production of H_2_S and its roles in pulmonary circulation. H_2_S-producing enzymes have been found to be expressed in the endothelium and the media of pulmonary arteries. CSE and 3-MST are detected in both PAECs and PASMCs, whereas CBS exhibits an endothelial predominance. H_2_S is involved in lung and PA development by (1) promoting vascular growth and associated PAEC migration and (2) decreasing media muscularization. Variation in the partial pressure of oxygen is linked to regulation of H_2_S metabolism. Hypoxia exposure triggers a decrease in H_2_S clearance and an increase in H_2_S production, leading to a global elevation of intracellular H_2_S levels. Modulation of the H_2_S clearance/production balance may play a pivotal role in oxygen sensing in pulmonary circulation. PA treatment with H_2_S donors is associated with paradoxical influence on vascular tone. On the one hand, H_2_S induces biphasic contraction of PAs that could be explained by the balance of the promotion (oxidation by SQR) versus poisoning (blocking of complex IV) effects of H_2_S on mitochondrial ETC function. In other hand, H_2_S also dose-dependently relaxes PAs through the activation of K_ATP_ channels, leading to vascular cells’ hyperpolarization and, thus, to relaxation. Crosstalk between H_2_S and endothelial NO pathways was also observed, suggesting a potential role of the endothelium in the relaxing effects of H_2_S. 3-MST: 3-mercaptopyruvate sulfur transferase; CSE: cystathionine γ-lyase; GYY4137: morpholin-4-ium 4-methoxyphenyl (morpholino) phosphinodithioate; H_2_S: hydrogen sulfide; K_ATP_: ATP-sensitive K^+^ channel; Na_2_S: sodium sulfide; NaHS: sodium hydrosulfide; NO: nitric oxide; PA: pulmonary artery; PAECs: pulmonary artery endothelial cells; PASMCs: pulmonary artery smooth muscle cells.

**Figure 4 cells-10-01477-f004:**
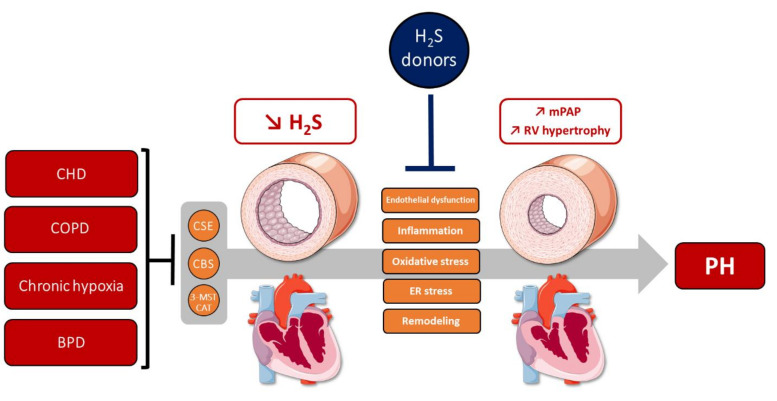
Alteration of H_2_S metabolism and protective influence of H_2_S donors on PH hallmarks. Clinical and experimental data suggest a pivotal dysregulation of endogenous H_2_S production through the inhibition of activity and/or expression of H_2_S-generating enzymes (namely, CSE, CBS, and 3-MST/CAT). These alterations promote the development of PH-associated hallmarks in PA, such as endothelial dysfunction, inflammation, and oxidative and ER stress, as well as increased media thickness of PAs. Altogether these pathological phenomena lead to increased PA resistance, mPAP, subsequent RV hypertrophy and, finally, PH, H_2_S donors such as NaHS and GYY4137 demonstrate multifaceted protective effects on PA alterations, counteracting PH development. 3-MST: 3-mercaptopyruvate sulfur transferase; BPD: bronchopulmonary dysplasia; CAT: cysteine aminotransferase; CBS: cystathionine β-synthase; CHDs: congenital heart diseases; COPD: chronic obstructive pulmonary disease; CSE: cystathionine γ-lyase; ER: endoplasmic reticulum; H_2_S: hydrogen sulfide; mPAP: mean pulmonary artery pressure; PA: pulmonary artery; PASMCs: pulmonary artery smooth muscle cells; PH: pulmonary hypertension; RV: right ventricle. Flat arrows represent inhibition interactions.

**Table 1 cells-10-01477-t001:** Summary of reported effects of the use of H_2_S-releasing molecules on various experimental PH models. α-SMA: α-smooth muscle cell actin; ATF6: activating transcription factor 6; BAL: bronchial alveolar lavage; CO: carbon monoxide; ET-1: endothelin-1; ER: endoplasmic reticulum; GYY4137: morpholin-4-ium 4-methoxyphenyl (morpholino) phosphinodithioate; Grp78: binding immunoglobulin protein; HO-1: heme oxygenase-1; ICAM-1: intercellular adhesion molecule-1; mPAP: mean pulmonary artery pressure; NaHS,: sodium hydrosulfide; PA: pulmonary artery; PAECs: pulmonary artery endothelial cells; PASMCs: pulmonary artery smooth muscle cells; RV: right ventricle; TNF-α: tumor necrosis factor-α; IL-6/8: interleukin 6/8; VE-cadherin: vascular endothelial-cadherin.

Experimental Models	H_2_S-Releasing Molecules	Animal Species	Summary of Reported Effects on PH	References
MCT	NaHS 56 μmol/kg/day (21 days)	Wistar rat	↘ mPAP ↘ RV hypertrophy ↘ PA remodeling ↘ ICAM-1, TNF-α, IL-6, IL-8 MCP-1 (plasma and lung)	[141]
	NaHS 56 μmol/kg/day (21 days)	Wistar rat	↘ mPAP ↘ TNF-α, IL-6 (lung)	[34]
	NaHS 1 mg/kg/day (7 days after MCT injection)	Sprague Dawley rat	↘ mPAP ↘ RV hypertrophy ↘ PA remodeling ↗ RV ejection fraction ↗ VE-cadherin, ↘ α-SMA (PA)	[50]
	ACS 14 46.5 mg/kg (7 days after MCT injection)	Sprague Dawley rat	↘ mPAP ↘ RV hypertrophy ↘ PA remodeling ↗ RV ejection fraction ↗ VE-cadherin, ↘ α-SMA (PA)	[56]
Hypoxia-induced PH	NaHS 14 μmol/kg/day (21 days)	Wistar rat	↘ mPAP ↘ RV hypertrophy ↘ PA remodeling	[35]
	NaHS 14 μmol/kg/day (21 days)	Wistar rat	↘ mPAP ↗ CO (Plasma) ↗ HO-1 (PA media)	[142]
	NaHS 14 μmol/kg/day (21 days)	Wistar rat	↘ mPAP ↘ number of muscularized PA ↘ collagen type I/III, elastin (PA media)	[143]
	NaHS 14 μmol/kg/day (21 days)	Wistar rat	↘ mPAP ↘ RV hypertrophy ↗ total antioxidant capacity (lung)	[53]
	NaHS 10 μmol/kg/day (21 days)	Broiler	↘ mPAP ↘ PA remodeling	[40]
	Garlic powder 100 mg/kg/day (5 days)	Sprague Dawley rat	↘ mPAP ↗ relaxation intralobar PA (90 min hypoxia)	[57]
	GYY4137 (concentration not reported, daily, 4 weeks)	Sprague Dawley rat	↘ mPAP, and PA resistances ↘ RV hypertrophy ↘ PA remodeling ↗ treadmill running distance ↘ RE stress proteins ATF6 and Grp78 (PA)	[51]
COPD models	NaHS 50 μmol/kg/day (12 or 24 weeks)	C57BL/6 mice	↘ mPAP ↘ RV hypertrophy ↘ TNF-α (BAL) ↘ 8-hydroxyguanine (lung)	[94]
	NaHS 56 μmol/kg/day (60 days)	Sprague Dawley rat	↘ apoptosis of PAEC	[150]
High pulmonary blood flow	NaHS 56 μmol/kg/day (11 weeks)	Sprague Dawley rat	↘ mPAP ↘ PA remodeling ↘ collagen type I/III (PA) ↘ ET-1 (lung)	[48]
	NaHS 56 μmol/kg/day (11 weeks)	Sprague Dawley rat	↘ mPAP ↗ PASMC apoptosis	[145]
BDP-PH	GYY4137 37.75 mg/kg/day (10 days)	Newborn rat pups	↘ PA remodeling ↘ RV afterload and hypertrophy ↗ pulmonary vessels density	[52]

## Data Availability

No new data were created or analyzed in this study. Data sharing is not applicable to this article.

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
