# Peer review of "Hydrogen Sulfide Metabolism and Pulmonary Hypertension"

_cells, 2021, doi:10.3390/cells10061477_

Round 1

Reviewer 1 Report

Overall, this comprehensive review article is well writing. 

A number of minor concerns are outlined below. 

1. Please add more wording on subheadings to help the reader easier understand and gain from each paragraph. For example, “2. H2S metabolism in the pulmonary vasculature, 2.1. Metabolism of H2S”, “2.1.1. Anabolism of H2S”, “2.1.2. Catabolism of H2S”, “2.2.1. CSE”, “2.2.2. CBS”, “2.2.3. 3-MST/CAT couple” are too short as well as in others subheadings.

2. There are some spelling and grammar errors in the manuscript. Please correct them.

Author Response

We thank the referee for his helpful comments.

Minor concerns

  1. We have added more wordings on subheadings for an easier understanding of the paragraphs.

  1. We carefully re-read the manuscript to correct spelling and grammar errors. One of our colleague (Pr Roger Marthan) also read the manuscript to correct such errors.

Reviewer 2 Report

The authors have presented a well written work describing the role of Hydrogen sulfide metabolism and PH.

Although the review is quite exhaustive, I felt that it was extremely Verbose.

The section 3.2 - Oxygen sensing and HPV is very diffused and lacks a clear focus and is extremely lengthy. The authors would have to condense this bi

The authors must condense all the sections in 4, keeping only the relevant things needed. The aim of a review is to gather the attention of the reader. Keeping it brief would bring it much attention and also gain visibility to the Journal. 

Author Response

We thank the referee for his helpful comments.

We have now focused the manuscript and only kept the main findings especially in sections 3.2 and 4. We removed the Verbose. One of our colleague, Pr Roger Mathan, a bilingual expert in lung pathophysiology, greatly helped us in this task therefore we added his name to the list of authors.

Round 2

Reviewer 2 Report

The authors have adequately addressed my questions